# Broadband Metasurface Absorber Based on an Optimal Combination of Copper Tiles and Chip Resistors

**DOI:** 10.3390/ma16072692

**Published:** 2023-03-28

**Authors:** Yongjune Kim, Jeong-Hae Lee

**Affiliations:** 1Department of Electrical Engineering, The University of Suwon, Hwaseong 18323, Republic of Korea; 2Department of Electronic and Electrical Engineering, Hongik University, Seoul 04066, Republic of Korea

**Keywords:** metasurface absorber, broadband, tiling, chip resistor, genetic algorithm

## Abstract

In this study, a broadband metasurface absorber composed of an optimal combination of copper tiles connected with four chip resistors is designed and experimentally verified. After fixing the locations of the chip resistors and setting their resistances to 100 Ω, the genetic algorithm (GA) is utilized to design the optimal copper tile pattern for broadband absorption. The optimal combination of the copper tiles is identified by determining the states of the square tile pairs between copper or air, depending on the one or zero states of the bit sequence created by GA, respectively. The full-wave simulation results of the optimized metasurface absorber confirmed a −10 dB reflectance bandwidth within the frequency range of 6.57 to 12.73 GHz for the normal incidence condition, with the fractional bandwidth being 63.83%. The accuracy of the metasurface absorber was verified through an experimental result that matched well with the full-wave simulated one.

## 1. Introduction

Electromagnetic (EM) metasurfaces are two-dimensional (2D) versions of metamaterials, i.e., engineered artificial materials that can realize unnatural constitutive parameters such as negative permittivity and/or permeability [1,2,3]. Based on their versatile functionalities, the metasurfaces have been adopted in many research topics such as transmissive [4,5] or reflective [6,7] antennas, cloaks [8,9], sensors [10,11], and absorbers [12,13,14,15,16,17,18,19,20,21,22,23,24,25,26,27,28,29,30]. Among them, EM metasurface absorbers have attracted great attention from the perspective of a wide range of possible applications such as sensing [14], energy harvesting [28], and stealth technologies [29].

An electromagnetic (EM) metasurface absorber is a kind of absorbing resonator of which the conductive pattern on the top side of a dielectric substrate is critically coupled with an incident EM wave [12,13]. Since its bottom side is blocked by a metallic sheet [14,15,16,17,18,19,20,21,22,23,24,25,26,27,28,29,30], a reflected EM wave ends up interfering with the incident one. By satisfying the constructive interference condition between the incident and reflected waves at the top surface, the electric current induced on the metapattern can be maximized [15,31]. For a metapattern composed of lossy material, such as carbon paste [15,16], indium tin oxide (ITO) [17,18], or silver nanowire [19], the incident EM wave is absorbed by the ohmic loss inside the pattern. When both the top metapattern and the substrate consist of a metal, such as copper, as well as a lossy substrate, such as flame retardant 4 (FR4) [20,21,22,23], the EM wave is dominantly dissipated by the dielectric loss. Since the electric (E) field inside the substrate is coupled with the maximized electric current induced on the metapattern, the dielectric loss can be boosted as long as the constructive interference condition is maintained.

Even though the metasurface absorbers described above have been successfully verified, each of them suffers from some inherent limitations. First, the metasurface absorber with a lossy metapattern must be printed on or combined with the top surface of the substrate [15,16,17,18,19]. In such cases, it may be difficult to integrate the metasurface absorber with other electronic components, such as microwave circuits or antennas. In contrast, the biggest merit of a metasurface absorber composed of a metal metapattern is that it can be easily fabricated with electronic circuits. For instance, such a metasurface absorber can be integrated into an antenna system to reduce back-radiation and radar cross-section [21], as well as to prevent any interference in the received spectrum induced by multiple reflections in the system [23].

However, the metasurface absorber utilizing a metal metapattern suffers from a narrow absorption bandwidth [20,21,22,23]. To overcome this limitation, metasurface absorbers connected with chip resistors have been proposed [24,25,26,27]. This configuration enables the extension of the absorption bandwidth by diminishing the quality factor Q of a series resistor-inductor-capacitor (RLC) circuit, which is an equivalent model of the metapattern. Although broad absorption bandwidths for such absorbers have already been reported, their adoption of typical shapes, such as rings [24,27], a fan [25], or eight arms [26], may suffer from complex optimization processes. A multitude of design parameters exist in this regard, be it in terms of the dimensions of the metal patches, or the locations and resistances of the chip resistors. Therefore, it requires time-consuming iterative simulations based on the researcher’s intuition and trial-and-error process to determine the optimal parameters. Considering these circumstances, it may be burdensome to design structures for various frequency bands or with different sizes of unit cells.

In this paper, a broadband metasurface absorber consisting of an optimal combination of square copper tiles connected with four chip resistors is designed and experimentally verified. The most significant advantage of the proposed metasurface absorber is that it can be designed via an automatic procedure by utilizing the genetic algorithm (GA) [15,16,22,23,32,33], which mimics the meiosis of a chromosome using a pair of two-bit sequences. After pixelating the top surface of the absorber with square tiles and fixing the resistance and location of the chip resistors, the copper-or-air states of the square tiles are determined automatically by matching them with the one- or zero states of a bit sequence optimally identified using GA. The proposed design scheme does not suffer from time-consuming massive simulation studies. Based on full-wave simulation results, a broad −10 dB reflectance bandwidth is confirmed in the 6.57 to 12.73 GHz range, with the fractional bandwidth being 63.83%. The absorbing performance is verified by measuring the reflectance using a system composed of transmitting-and-receiving antennas.

## 2. Optimization of Copper Tiles

To identify the optimal combination of copper tiles, the top surface of an FR4 substrate is pixelated after fixing the locations of the four chip resistors, as shown in Figure 1a. The relative permittivity and the loss tangent of FR4 are 4.46 and 0.029, respectively. The size of the square unit cell in Figure 1a is set to 11.5 mm, along with an electrical length of 0.38$\lambda_{0}$, where $\lambda_{0}$ indicates the free-space wavelength at 10 GHz. A unit cell size less than 0.5$\lambda_{0}$ can prevent the grating lobe problem [34], which could be generated by the scattered electromagnetic waves from the metapattern. At the outer boundary of the 22 × 22 tiles, an air gap of 0.25 mm width is maintained to separate the unit cell from the neighboring ones. The size of the square tile is determined as 0.5 mm to match that of the 0402 type chip resistor having a width and length of 0.5 and 1 mm, respectively. To reduce simulation resources, such as memory and the computational running time, the transition boundary condition of Comsol Multiphysics, one of the commercialized finite-element-method (FEM) tools, is applied to the 2D tiles. The option enables approximated an analysis of the three-dimensional (3D) conductive layer using 2D one, by setting the conductivity and thickness of the layer [35]. For the *n*th pair of the square tiles, the conductivity is set to $\beta_{n}\times \sigma$, where $\beta_{n}$ and σ indicate the *n*th bit, i.e., 1 or 0, included in the bit sequence and the conductivity of the copper $5.8\times 10^{7}$ S/m, respectively. The thickness of the tile layer is set to 0.0348 mm.

To simulate the chip resistors, presented as 2D red squares in Figure 1, the lumped element of the simulator is used to set their resistances. The size of the red square is set to 0.5 mm under the assumption that the electrode of the chip is soldered with rectangular copper patches, indicated in green in Figure 1. The length and width of the rectangular copper patch are 1.5 and 0.5 mm, respectively. To guarantee the availability of sufficient area for soldering, an additional square copper patch, denoted in green in Figure 1, is attached to the other side of the rectangular patch. The locations of the center of the chips are (x,y)=(±3.25mm,0) and (0,±3.25mm). The four chip resistors guarantee the symmetry of the metapattern for the *x* and *y* axes simultaneously. To maintain the absorbing functionality of the metasurface absorber independent for the horizontal incident angle ϕ, the symmetry of the pattern should be achieved [15,16,23]. The locations of the chips are slightly shifted outward from the centers of the quadrants (x,y)=(±2.75mm,0) and (0,±2.75mm) to utilize as many tiles located near the boundary as possible. Even though the centers are slightly shifted toward the outer boundary of the unit cell, the rectangular patches connected to the chips enable searching for the combination of copper tiles that not only diverge from, but also converge to the center of the unit cell.

The square tiles marked in blue in Figure 1b–f indicate the first, second, third, fourth, and sixty-second pairs that are matched with each bit in the sequence. To guarantee polarization independence for the normal incidence of the EM wave, the pairs are grouped to maintain the axial symmetries for both the *x* and *y* axes. When the bit is one or zero, the pair is set to copper or air, respectively. Figure 1g shows the optimal combination of copper tiles as identified using GA, of which the detailed procedure will be described in the next paragraph. Contrary to the expectation for the effect of the shifted location of the chip resisters, it is found that the pattern is converged into the center of the unit cell, as indicated by orange arrows in Figure 1g. The automatic design by GA is enabled by setting the shape of the copper pad for the soldering process to a T-shaped one denoted in green in Figure 1, which enables searching for the combination of copper tiles that not only diverge from, but also converge to the center of the unit cell. Based on the result, it could be expected that the optimal metapattern can be found while the locations of the chip resistors are determined near the center of the quadrants. Figure 1h shows the distribution of the electric current density on the metapattern at 10 GHz.

Figure 1i depicts the simulation settings for the unit cell of the metasurface absorber. As shown in Figure 1i, an air box of height 0.5$\lambda_{0}$ is attached on top of the metasurface and is truncated by a perfectly matched layer of height 0.1$\lambda_{0}$. To enable the design in an infinite array condition, the periodic boundary condition is utilized at the side boundaries of the simulation domain. Furthermore, the port is assigned at the top surface of the air box, which irradiates the plane wave with an electric field intensity of 1 V/m. The bottom side of the FR4 is blocked by the perfect conductor. The thickness of FR4 is set to 3.6 mm, with an electrical length of 0.25$\lambda_{g}$, where $\lambda_{g}$ indicates the wavelength in FR4 at 10 GHz. This thickness is specially chosen to achieve constructive interference between the incident and reflected EM waves at the surface where the metapattern is located [15]. In addition, the thickness of $\lambda_{g}/4$ enables the impedance matching condition to be approximately achieved when the sheet resistance of the metapattern is designed at around 377 Ω [36].

Since the total number of sequences is 62, the same number of bits are included in a sequence. As an initial condition for GA, a pair of two-bit sequences is selected from the randomly generated 62 sequences. For stochastically searching an elite seed pair, the same number of samples as the number of bits could be regarded as sufficient [15,16,22,23,33]. After competing a figure of merits (FOMs) of all the sequences, two of them are selected as the elite pair, of which FOMs are minimum. The FOM is defined as follows:(1)$$\mathrm{FOM}=\frac{1}{9}\sum_{i=1}^{3}R_{TE,\theta =0^{{}^{\circ}},f_{i}}+R_{TE,\theta =60^{{}^{\circ}},f_{i}}+R_{TM,\theta =60^{{}^{\circ}},f_{i}},$$
where $R_{TE,\theta =0^{{}^{\circ}},f_{i}}$ indicates the reflectance of transverse electric (TE) polarization at an incident angle of $0^{{}^{\circ}}$, while $R_{TE,\theta =60^{{}^{\circ}},f_{i}}$ and $R_{TM,\theta =60^{{}^{\circ}},f_{i}}$ are those of the TE and transverse magnetic (TM) polarizations at an incident angle of $60^{{}^{\circ}}$, respectively. The *i*th frequency is indicated by $f_{i}$ when the first, second, and third frequencies are 8, 10, and 12 GHz, respectively. The TE and TM polarizations are defined by the directions of the electric and magnetic fields, respectively, of the incident EM waves that are perpendicular to the plane of incidence. In this case, the reflectance of TM polarization for the normal incidence is omitted because the response is the same as that for TE polarization, owing to the axial symmetries of the metapattern.

After selecting two bit sequences, the conventional GA is applied to them. First, cross-operation is performed 16 times, leading to exchanges in the bit sequence between the pair above 16 selected random positions. At this stage, new 62-bit sequences are generated by repeatedly applying the same operation on the former elite pair. Following this, the mutation operation is applied to five bits included in each sequence, which reverses one to zero, or vice versa. Next, the new elite pair of bit sequences is selected by implementing the same competing procedure. Because GA is a kind of stochastic optimizers, it is difficult to set the optimal FOM in advance. Therefore, the entire procedure described above is then repeated 15 times until the FOMs of two sequences converge sufficiently [15,16], as shown in Figure 1j. Then, the first sequence of the 12th iteration in Figure 1j is selected as the best one by checking the −10 dB reflectance bandwidth.

To confirm whether a resistance of 100 Ω is the best choice, the same design process described above is applied, considering resistances of 80.6, 90.9, 110, and 120 Ω, which are the standard resistances of commercialized chip resistors. To maintain the axial symmetries of the metasurface absorber for both the *x* and *y* axes, all four resistors are set to produce the same resistance for each case. By utilizing the same setting as shown in Figure 1 and applying GA to it under the same conditions except for the resistance, the optimal absorbing performance for each case is identified for the normal incidence of EM wave with TE polarization, as shown in Figure 2a. From Figure 2a, the −10 dB reflectance bandwidths are confirmed to be in the ranges of 6.69–12.54, 7.02–12.52, 6.48–12.68, 6.55–12.56, and 6.21–12.36 GHz for 80.6, 90.9, 100, 110, and 120 Ω, respectively. Among these, the widest −10 dB bandwidth is confirmed as 6.2 GHz for a resistance of 100 Ω, which is calculated by subtracting the highest and lowest frequencies of the band. Meanwhile, its fractional bandwidth is calculated to be 64.72% on dividing the bandwidth by the center frequency of the band. Although the fractional bandwidth in the case of 120 Ω is confirmed as 66.24%, which is slightly wider than that for the 100 Ω case, the case presenting the widest absolute bandwidth is chosen in this study.

Although the design strategy implemented in this study is similar to that of the pixelated metasurface absorber combined with resistive loads [28], the latter focused on achieving a dual-band absorption rather than a broad one by considering the resistance of the loads as the input impedance of the harvesting system. Moreover, the proposed configuration of copper tiles connected with resistors on the surface is quite different from that of the dual-band pixelated metasurface absorber, in which resistors are connected to the slotted ground through via-holes [28]. Furthermore, the binary particle swarm optimization algorithm used in the previous study [28] is different from the GA utilized in the proposed study.

Figure 2b shows the absorption *A* of the optimal design using 100 Ω, which is calculated by subtracting the reflectance *R* from the normalized total power 1, i.e., A=1−R, for the normal incidence. The reflectance is defined by the ratio of the total irradiated power from the port and the specularly reflected power from the metasurface absorber. To confirm whether the reduction in reflectance guarantees a precise evaluation of absorption, another result that includes the effect of scattering by the metapattern is compared to that of the process defined above. The result is calculated as:(2)$$A=1-R-\frac{P_{scattered}}{P_{in}},$$
where $P_{scattered}$ and $P_{in}$ indicate the integrated power of the EM wave propagating outward through the side boundaries of the simulation domain shown in Figure 1g and the total input power, respectively. From the comparison in Figure 2b, it is confirmed that the effect of scattering is minor, and the reduction in reflectance primarily originates from the absorption.

Since FOM is set to include the reflectance for the oblique incidences at an elevation angle θ of $60^{{}^{\circ}}$ for the TE and TM polarizations, it can be expected that their −10 dB reflectance bandwidths could be achieved as wide as that of the normal incidence. However, it is confirmed that the bandwidths for the cases considered in this study are not sufficiently attained in the target frequency band, as shown in Figure 2c. Although the absorbing performances are not perfectly achieved for the $\theta =45^{{}^{\circ}}$ and $60^{{}^{\circ}}$ oblique incidence cases, the broadband performance is maintained from 6.79 to 11.33 GHz, with a fractional bandwidth of 50.1% for the oblique incidences up to $\theta =30^{{}^{\circ}}$ for both the TE and TM polarizations, as shown in Figure 2d. To verify the accuracy of the simulation using Comsol Multiphysics, the same configuration as Figure 1i is considered while utilizing the optimal metapattern in Figure 1g for simulation using the Ansys high-frequency structure simulator (HFSS). In addition, 3D copper tiles are utilized for both simulations of Figure 2c,d instead of the 2D ones. By utilizing the 3D pattern, the −10 dB reflectance bandwidth is confirmed to be in the range of 6.57 to 12.73 GHz, with a fractional bandwidth of 63.83% for the normal incidence. The results confirm the lack of any noticeable difference between them, as presented in the comparisons in Figure 2c,d.

Table 1 shows the comparisons of the simulated −10 dB reflectance bandwidths of the proposed metasurface absorber with those of the single-layer absorbers utilizing chip resistors. From the comparisons of the fractional bandwidths for the normal incidence, it is confirmed that the proposed result is approximately equal to that of the fan [25] and wider than that of the eight arms-shaped absorbers [26]. Meanwhile, although the bandwidth is slightly narrower than that of the double ring absorber [27], the bandwidth of the proposed absorber for the $30^{{}^{\circ}}$ oblique incidence is widest among all of the metasurface absorbers summarized in Table 1. Furthermore, based on the proposed metapattern, a 7.88% fractional bandwidth is achieved for the $60^{{}^{\circ}}$ oblique incidence, which is not applicable for the fan [25] and the double ring [27] absorbers. The electrical thickness of the proposed metasurface absorber for the center frequency of the normal incidence case is slightly thicker than that of the others, but the resulting differences are not observed to be significant. Furthermore, thicknesses are approximately retained around the quarter of the wavelength.

## 3. Experimental Verifications

To verify the absorbing performances experimentally, the proposed metasurface absorber is fabricated, as shown in Figure 3a. It is fabricated based on the conventional surface mount technology [37]. Therefore, the cost is moderate as that required for printed-circuit-board (PCB) fabrication. The dimension of the sample is 150×150${\mathrm{mm}}^{2}$, composed of 12×12 unit cells. The copper metapattern combined with the chip resistors is fabricated on a 2 mm thickness FR4 substrate. To restrict the thickness of the total structure to 3.6 mm, another 1.6 mm thickness FR4 substrate is attached to the former one, while the bottom side is covered by a copper sheet. Figure 3b shows the measured thickness of the fabricated sample. An insulating green film is used for the soldering process, covering the top surface of the pattern, except for the area containing the chip resistors. To guarantee electrical connectivity among the copper tiles, a short patch of 0.1 mm width is inserted between the adjacent corners of two copper tiles connected through each vertex. The complemented unit cell is presented in Figure 3c. Figure 3c is designed using the Ansys HFSS, which guarantees the same accuracy as Comsol Multiphysics, as verified in Figure 2c,d. A comparison is conducted between the simulation results of the complemented design and those of the original design, considering the normal and $15^{{}^{\circ}}$ oblique incidences, respectively, as shown in Figure 3g,h. The results confirm that no performance degradations occur upon adding short copper patches at the connecting points.

Figure 3d shows the fabricated unit cell enlarged from Figure 3a. The white lines surrounding the chip resistors indicate their locations for the soldering process. To measure the reflectance of the fabricated metasurface absorber, the measurement settings in Figure 3e,f are used to estimate the normal or oblique incidences, which utilize one or two X-band horn antennas, respectively. The distance between the antenna and the sample is determined to be 75 mm, which satisfies at least the far-field condition of the incident EM wave from an antenna with aperture dimensions of 78×57${\mathrm{mm}}^{2}$. Since the sample size is relatively small, the minimum distance is considered as being advantageous for accurately irradiating the EM wave on the sample. The backside of the sample is truncated by using a blue-pyramid-shaped microwave absorber and the edge of the sample holder; as well, the upper surface of the rotating arms are coated by a black-flat microwave absorber to prevent unintended scattering. First, the reflectance of the sample for the normal incidence is measured using ${\left|S11\right|}^{2}$, which indicates the ratio between the transmitted and the received reflected power through the antenna connected to port 1. After this, the reflectance of the copper plate with the same dimensions as the sample, is measured. To eliminate unintended multiple reflections, the Hanning window is multiplied to the main pulse reflected by the target in the time domain using the Anritsu vector network analyzer (MS46522B). By subtracting the reflectance of the copper plate from that of the metasurface absorber in the decibel (dB) scale in the frequency domain, the normalized reflectance shown in Figure 3g can be estimated. For the simulation, the reflectance calculated by the setting in Figure 1g is the same as the normalized one because the reference reflectance is 0 dB.

Figure 3g confirms that the measured result using the TE polarization matches well with the simulation results of the original and complemented unit cells using the same polarization. The simulations for the TM polarization are omitted because the results are the same as those in the case of TE, owing to the axial symmetries of the metapattern. On the other hand, a slight discrepancy is observed between the measured results using the TE and TM polarizations due to the different antenna-beam patterns for the fixed measurement environment. For the TM polarization, the antenna in Figure 3e is rotated about $90^{{}^{\circ}}$ along the normal direction of the antenna aperture. Since the background of the measurement environment and the sample holder are not perfectly covered by the blue-pyramid and black-flat EM absorbers, respectively, unintended scattering might also exist possibly in different ways for the two antenna settings. Despite these discrepancies, the measured results approximately matched the simulated ones, thus verifying the performance of the proposed metasurface absorber.

To verify the reduction in normalized reflectance for the oblique incidence, the reflectance from the sample is measured through ${\left|S21\right|}^{2}$, which reflects the ratio between the transmitted and the received reflected power through ports 1 and 2, respectively. Both ports are connected to the same X-band horn antennas, as shown in Figure 3f. The normalized reflectance for the oblique incidence is estimated using the same method as for the normal incidence. Figure 3h shows the simulated and measured results for an incident angle $\theta =15^{{}^{\circ}}$ using both the TE and TM polarizations. Although differences are observed between the measured and simulated results, the absorbing performances of the proposed metasurface absorber can be verified at levels below −10 dB for both types of polarizations. These differences may have originated as a result of direct coupling between antennas [38] or the relatively short distance between the sample and the transmitting antenna. Since there exists a beam angle covering the area of the sample, errors in incident angles are inevitable. They may be increased for unit cells located far from the center of the sample. Moreover, the deterioration of the absorbing performance is observed to be more critical at the steeply inclined incidence; therefore, the experimental verification is confined with $\theta =15^{{}^{\circ}}$. The error could be mitigated by adopting the measurement system using two lens-horn antennas, which can collimate the incident beam to be irradiated on the sample [16,39].

## 4. Conclusions

This study proposes a broadband metasurface absorber consisting of an optimal combination of copper tiles connected with chip resistors. The proposed metasurface absorber is designed utilizing GA, of which FOM includes the reflectance of the normal and $60^{{}^{\circ}}$ oblique incidences with the TE and TM polarizations. The full-wave simulation results, reached by utilizing the 3D copper metapattern, confirmed a −10 dB reflectance bandwidth in the frequency range of 6.57 to 12.73 GHz, with a fractional bandwidth of 63.83% for the normal incidence. The broadband absorption performance is experimentally verified by measuring the reflectance of the fabricated sample, normalized by that of the reference copper plate. Moreover, it is confirmed that broadband performance is maintained from 6.79 to 11.33 GHz, with a fractional bandwidth of 50.1% for oblique incidences up to the angle θ of $30^{{}^{\circ}}$ with both the TE and TM polarizations. Although the absorbing bandwidth for the $30^{{}^{\circ}}$ incident angle is confirmed as the widest one by comparing it with those of the other chip resistor based metasurface absorber [25,26,27], there is still room for improvement by adopting the design strategy utilizing the deep-learning algorithm [40,41] which may enable the optimization of not only the copper pattern, but also the locations of the chip resistors at once. The metasurface absorber composed of optimal copper tiles soldered with chip resistors has a great merit in that they can be easily fabricated with electronic circuits. This may contribute to the development a novel technique for designing antenna systems in which interferences among antennas resulting from multiple reflections or mutual coupling in a broad bandwidth could be eliminated.

## Figures and Tables

**Figure 1 materials-16-02692-f001:**
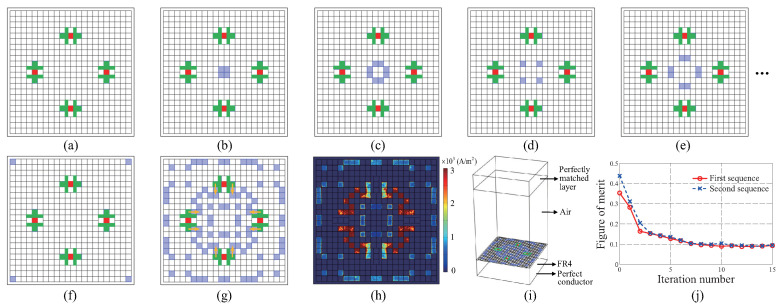
Schematics of the pixelated top surface of the proposed metasurface absorber: (**a**) Default setting of the chip resistors (red) and the patches (green) for soldering the chips. (**b**) The first pair of square tiles. (**c**) Second pair of square tiles. (**d**) Third pair of square tiles. (**e**) Fourth pair of square tiles. (**f**) Sixty-second pair of square tiles. The conductivity of the *n*th pair is set to $\beta_{n}\times \sigma$, where $\beta_{n}$ and σ indicate the nth bit, i.e., 1 or 0, included in the bit sequence and the conductivity of the copper $5.8\times 10^{7}$ S/m, respectively. (**g**) The optimized metapattern: Blue represents copper and white refers to air. Orange arrows: Directions of expansion of the copper patches. (**h**) Distribution of the electric current density on the metapattern. (**i**) Simulation setting for full-wave simulation: The top of the air box is set to port, while all the side boundaries are set to the periodic boundary condition. (**j**) Convergences of the figure of merits (FOMs) of a pair of two elite sequences.

**Figure 2 materials-16-02692-f002:**
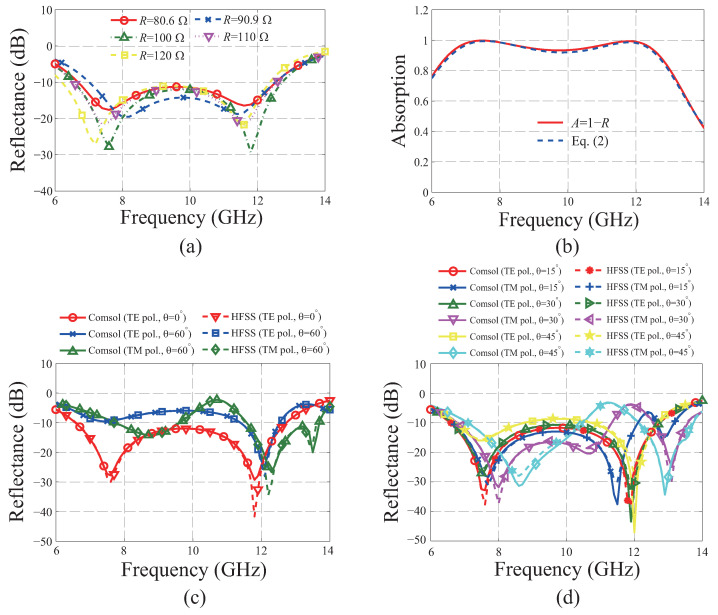
Full-wave simulation results of the proposed metasurface absorber based on 100 Ω resistance: (**a**) Comparison of the reflectance of the optimal design with those having 80.6, 90.9, 110, and 120 Ω for the normal incidence of electromagnetic (EM) wave with transverse electric (TE) polarization. (**b**) Calculated absorptions for the normal incidence before and after considering the power scattered outward through the side boundaries of the simulation domain, as shown in Figure 1i. (**c**) Reflectance for both TE and transverse magnetic (TM) polarizations with incident angles of $0^{{}^{\circ}}$ and $60^{{}^{\circ}}$. (**d**) Reflectance for both TE and TM polarizations with incident angles of $15^{{}^{\circ}}$, $30^{{}^{\circ}}$, and $45^{{}^{\circ}}$.

**Figure 3 materials-16-02692-f003:**
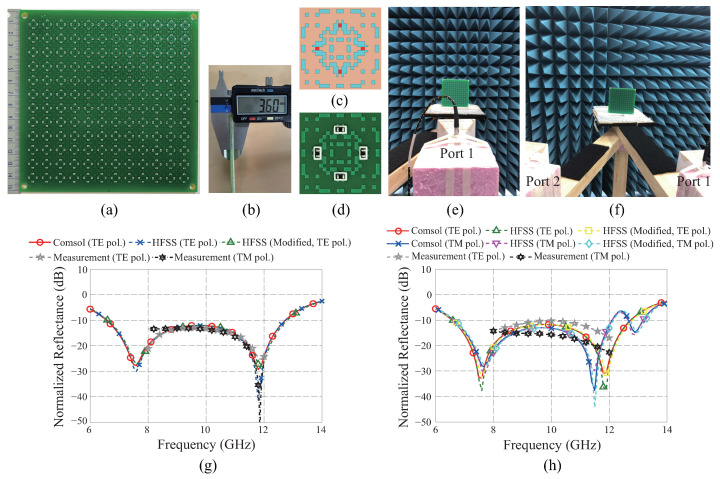
Experimental verification of the proposed metasurface absorber: (**a**) Fabricated 12×12 unit cells. (**b**) Measured thickness of 3.6 mm. (**c**) Schematic of the complemented unit cell where adjacent corners of the square pixels are connected by short copper patches. (**d**) The fabricated unit cell. Measurement setting for the TE polarizations: (**e**) Normal and (**f**) $15^{{}^{\circ}}$ oblique incidences. Comparison of simulated and measured reflectance for an elevation angle of $15^{{}^{\circ}}$ using both TE and TM polarizations: (**g**) Normal incidence and (**h**) Oblique incidences.

**Table 1 materials-16-02692-t001:** Comparisons of the simulated −10 dB reflectance bandwidths (BW) of single-layer metasurface absorbers combined with chip resistors for the normal and oblique incidences. For the oblique incidences, BWs are defined as the intersected ranges of BWs of the TE and TM polarizations for each case. The incident angle and the wavelength in the substrate at the center frequency of the normal incidence case are indicated by θ and $\lambda_{g}$, respectively. NA: Not applicable.

	Normal Incidence	Oblique Incidences		Thickness of
	Fractional BW (%)	θ	Fractional BW (%)	Substrate ($\lambda_{g}$)
Figure 2 of [25]	63.41 (7–13.5 GHz)	$30^{{}^{\circ}}$	34.59 (7.34–10.41 GHz)	0.23 (3.4 mm)
		$60^{{}^{\circ}}$	NA	
Table 1 of [26]	48.85 (8.2–13.5 GHz)	$30^{{}^{\circ}}$	41.55 (8.2–12.5 GHz)	0.21 (3 mm)
		$60^{{}^{\circ}}$	11.76 (10.4–11.7 GHz)	
Figure 5 of [27]	67.69 (8.6–17.4 GHz)	$30^{{}^{\circ}}$	7.94 (13.3–14.4 GHz)	0.22 (2.4 mm)
		$60^{{}^{\circ}}$	NA	
This work	63.83 (6.57–12.73 GHz)	$30^{{}^{\circ}}$	50.1 (6.79–11.33 GHz)	0.25 (3.6 mm )
		$60^{{}^{\circ}}$	7.88 (11.58–12.53 GHz)	

## Data Availability

Not applicable.

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
