# Peer review of "Broadband Metasurface Absorber Based on an Optimal Combination of Copper Tiles and Chip Resistors"

_materials, 2023, doi:10.3390/ma16072692_

Round 1
Reviewer 1 Report
In this manuscript, a broadband metasurface absorber composed of an optimal combination 2 of copper tiles connected with four chip resistors is designed and experimentally verified. The genetic algorithm 4 (GA) is utilized to design the optimal copper tile pattern for broadband absorption. Overall, the idea is interesting but some revisions are still suggested before the acceptance.
1. The major question is that why the four chip resistors are used and located at the (±3.25 mm, 0). How about other numbers and other places? The reviewer think this point should be clarified. I found the authors have studied the change of the resistance to evaluate the best one, however the place I think will also have an influence on the overall absorption.
2. In addition, the author may also clarify how to determine the thickness of the meta-atom and the period.
3. Although the pattern was optimized using GA algorithm, it is still suggested to analyze the role of each part of the pattern. For example, how about the performance if it is without the outer pixels.
4. More advanced method using deep-learning-assisted method [e.g., Advanced Photonics Nexus 2.1 (2023): 016010.] maybe useful for the further designs.
Reviewer 2 Report
In this manuscript, the authors proposed a metasurface absorber consisting of an optimal combination of copper tiles with four chip resistors. The pattern of copper tiles was searched by genetic algorithm for broadband absorption. Moreover, such a design was validated in experiments. The topic is interesting and the manuscript is well organized. The results are sound and solid. I would like to recommend its publication after addressing following issues:
1) I suggest authors strengthen the motivation of this work and highlight the advances in the introduction part. The current description is not persuasive enough.
2) Figure 1(b) to 1(e) is confusing in my opinion. I understand that the authors made some description in the context, but still I suggest to make it more clear in the caption at least. Meanwhile, is there any other way to present such a relation? Because only providing figures for 1st, 2nd,3rd, and jumped suddenly to 62nd pair really made it confusing.
3) The authors should provide detailed information of “transition boundary condition” in comsol. Moreover, other simulation details should be added.
4) What is the optimal (expected best) value for the FOM? Did the GA successfully make the structure approaching to it?
5) I’m lost in the modified structure shown in Figure 3(c)? Why the design was modified? To make comparison?
Reviewer 3 Report
Please find the attachmenet

Reviewer 4 Report
This work presents an efficient wideband absorber metasurface by combining copper tiles and chip resistors.
My observation is that the paper is well-structured and the results look convincing. The work has also an acceptable level of novelty, however it is not well highlighted. While I found this work interesting and informative for the community, there are some aspects that certainly need improvements before making the final call on this submission. Here are the detailed comments:
Please explain the boundary condition used in Fig.1. and elaborate on the simulation setting.
The novelty is not well explained in the paper.
There are some discrepancies between the measured and simulated results in Fig. 3 why? Please clarify how these errors can be mitigated.
The introduction is a bit abrupt and does not adequately umbrella the state-of-the-art in relation to metasurfaces in both aspects of manufacturing and application. A more comprehensive introduction about metasurface and their applications are necessary. It should be mentioned that metasurafcaes have been used to develop antenna-senor antenna, as explained in: “Tunable terahertz filter/antenna-sensor using graphene-based metamaterials, 2022. Electromagnetic filtering is another important application of metasurfaces as explained in: All-metal wideband frequency-selective surface bandpass filter for TE and TM polarizations. Metasurfaces have effectively been used for beamsteering as explained in: Beam‐steering of microstrip antenna using single‐layer FSS based phase‐shifting surface. Another important application of metasurafce is transmit array as explained in All-metal wideband metasurface for near-field transformation of medium-to-high gain electromagnetic sources.
Some of the recent significant improvements in environmental sensing have been achieved through metasurface concept as explained in: advancements and artificial intelligence approaches in antennas for environmental sensing.
What about the measurement setting? Elaborate so it can be replicable.
Please make comments on the fabrication cost of the proposed metasurface. Is it considered low-cost? How?
Results and discussion: Please capture filed (or current distribution) of the metasurafce at critical frequencies .
One important aspect missing in this paper is about manufacturing mrtasurfaces. Almost all metasurafces mentioned in this paper are printed metasurfaces. While metasurfaces can be all-dielectric and all-metal. In terms of metasurface manufacturing and prototyping, it needs to be mentioned that metasurfcaes can be made of all-dielectric material. The following reference is recommended for this purpose.
M. Norouzi, et al “3D metamaterial ultra-wideband absorber for curved surface”, Scientific Reports, 2023.
Round 2
Reviewer 4 Report
All concerns are addressed